# The impact of stopping and starting indoor residual spraying on malaria burden in Uganda

Jane F. Namuganga[1,10], Adrienne Epstein [2,10 ✉], Joaniter I. Nankabirwa[1,3], Arthur Mpimbaza[1,4], Moses Kiggundu[1], Asadu Sserwanga[1], James Kapisi[1], Emmanuel Arinaitwe[1], Samuel Gonahasa[1] Jimmy Opigo[5], Chris Ebong[1], Sarah G. Staedke[6], Josephat Shililu[7], Michael Okia[7], Damian Rutazaana[5], Catherine Maiteki-Sebuguzi[5], Kassahun Belay[8], Moses R. Kamya[1,3], Grant Dorsey[9] & Isabel Rodriguez-Barraquer[9]

The scale-up of malaria control efforts has led to marked reductions in malaria burden over the past twenty years, but progress has slowed. Implementation of indoor residual spraying (IRS) of insecticide, a proven vector control intervention, has been limited and difficult to sustain partly because questions remain on its added impact over widely accepted interventions such as bed nets. Using data from 14 enhanced surveillance health facilities in Uganda, a country with high bed net coverage yet high malaria burden, we estimate the impact of starting and stopping IRS on changes in malaria incidence. We show that stopping IRS was associated with a 5-fold increase in malaria incidence within 10 months, but reinstating IRS was associated with an over 5-fold decrease within 8 months. In areas where IRS was initiated and sustained, malaria incidence dropped by 85% after year 4. IRS could play a critical role in achieving global malaria targets, particularly in areas where progress has stalled.

[1] Infectious Diseases Research Collaboration, Kampala, Uganda. [2] Department of Epidemiology and Biostatistics, University of California San Francisco, San Francisco, CA, USA. [3] Department of Medicine, Makerere University, College of Health Sciences, Kampala, Uganda. [4] Child Health and Development Centre, Makerere University, College of Health Sciences, Kampala, Uganda. [5] National Malaria Control Division, Ministry of Health, Kampala, Uganda. [6] London School of Hygiene and Tropical Medicine, London, UK. [7] US President's Malaria Initiative – VectorLink Uganda Project, Kampala, Uganda. [8] US President's Malaria Initiative, USAID/Uganda Senior Malaria Advisor, Kampala, Uganda. [9] Department of Medicine, University of California San Francisco, San Francisco, CA, USA. [10] These authors contributed equally: Jane F. Namuganga, Adrienne Epstein. ✉email: Adrienne.Epstein@ucsf.edu

Over the past 20 years the scale-up of malaria control efforts has led to marked reductions in morbidity and mortality[1,2]. However, global progress has slowed in recent years, particularly in sub-Saharan Africa, which accounted for 94% of the world's 215 million cases in 2019[2]. Long-lasting insecticidal nets (LLINs) and indoor residual spraying (IRS) of insecticide are the primary vector control interventions used for the prevention of malaria. The World Health Organization recommends universal coverage of LLINs for at-risk populations in sub-Saharan Africa, where the proportion of households owning at least one LLIN is estimated to have increased from 47% in 2010 to 72% in 2018. Pyrethroids are the only class of insecticides widely use in LLINs and, given the emergence of widespread pyrethroid resistance[3,4], there is concern that the effectiveness of LLINs may be diminishing, leading to the development of new LLIN formulations including pyrethroid synergists and non-pyrethroid nets. Unlike LLINs, IRS has the advantage of utilizing multiple different classes of insecticides and combining IRS with LLINs may improve malaria control and slow the spread of pyrethroid resistance. However, few controlled trials have evaluated the effect of adding IRS to communities using LLINs and the evidence is mixed, with a few studies showing benefits when IRS included "non-pyrethroid-like" insecticides[5]. Other barriers to IRS delivery—including cost, logistics, and community acceptance—have limited its use[6], such that less than 2% of the population at risk in sub-Saharan Africa was protected by IRS in 2019, a decrease from over 10% coverage in 2010[2].

Uganda is illustrative of a country where the burden of malaria remains high and progress has slowed in recent years[2]. Malaria control efforts in Uganda have primarily focused on LLINs. In 2013–14 it became the first country to implement a universal LLIN distribution campaign, which was repeated in 2017–18. In 2018–19, Uganda had the highest coverage of LLINs in the world, with 83% of households reporting owning at least one LLIN[7]. In contrast to LLINs, the implementation of IRS in Uganda has been focal and limited. In 2006, IRS was reintroduced into Uganda for the first time since the 1960s. In 2007–09, the IRS program was shifted to ten high burden districts in the north, leading to large reductions in malaria burden[8,9]. In 2014, the IRS program was relocated from these ten northern districts to 14 districts in the eastern part of the country, where it has been sustained. The discontinuation of IRS in the ten northern districts was followed by a marked resurgence in malaria cases[10,11], prompting the implementation of a single round of IRS in these ten districts in 2017.

In this study, we use data from a network of health facility-based malaria surveillance sites to evaluate the impact of different IRS delivery scenarios in 14 districts in Uganda. This study has three objectives: (1) to estimate the impact of withdrawing IRS after 5 years of sustained use on the burden of malaria in three sites in Northern Uganda; (2) to estimate the impact of restarting IRS with a single round 3 to 4 years after IRS was discontinued on the burden of malaria in nine sites in Northern Uganda; and (3) to estimate the impact of 5 years of sustained IRS on the burden of malaria in five sites in Northern and Eastern Uganda.

## Results

**Study sites and vector control interventions**. This study utilized data from 14 health facilities located in 14 districts in Northern and Eastern Uganda (Fig. 1) which were part of a larger comprehensive malaria surveillance network called the Uganda Malaria Surveillance Program (UMSP). Between 2007 and 2009, IRS was implemented in ten high burden districts in northern Uganda[12]. DDT or pyrethroids were initially used but in 2010 the

insecticide was changed to a carbamate (bendiocarb) due to concern regarding the spread of pyrethroid resistance. Rounds of bendiocarb were repeated approximately every 6 months until 2014 when the IRS program was discontinued, so that resources could be shifted to other high burden districts. In 2017, these ten districts in northern Uganda received a single round of the organophosphate pirimiphos-methyl (Actellic 300CS®) following reports of malaria resurgence after IRS has been discontinued in 2014. Between 2014 and 2015, IRS with bendiocarb was implemented in 14 districts in the Northern and Eastern part of the country. Rounds of bendiocarb were repeated approximately every 6 months until 2016 when the formulation was changed to Actellic 300CS®, which continues to be administered once a year.

Universal LLIN distribution campaigns were conducted in 2013–14 and 2017–18, where LLINs were distributed free-of-charge by the Uganda Ministry of Health targeting one LLIN for every two household residents. In 2013–14, all districts across the country received LLINs impregnated with pyrethroid insecticides. In 2017–18, the Ministry of Health distributed both conventional LLINs and LLINs containing piperonyl butoxide (PBO) due to concerns of pyrethroid resistance. During the latter distribution, all districts included in this analysis received conventional pyrethroid insecticides due to prior concerns of antagonism between PBO LLINs and Actellic 300CS®.[13]

We assessed the impact of stopping IRS after sustained use (objective 1), reinitiating IRS for a single round after stopping 3 years prior (objective 2), and initiating and sustaining IRS in areas that had not received the intervention prior (objective 3) on changes in malaria case counts relative to a baseline before to starting or stopping IRS. Figure 2 shows the timeline of control interventions and study timelines (with more detail for each site in Supplementary Fig. 1).

**Impact of withdrawing IRS after sustained use**. Across the three sites included in the analysis, a total of 224,859 outpatient visits were observed (Table 1). During the baseline period, average monthly cases ranged from 104–272 and test positivity rate (TPR), the proportion of individuals tested for malaria that resulted in a positive test, ranged from 23.7–25.9%. This increased to 491–751 and 52.3–78.0% respectively, during the evaluation period (Supplementary Fig. 2).

Monthly adjusted IRRs and 95% confidence intervals (CI) for the three sites combined are presented in Fig. 3 and Supplementary Table 1. These results showed an initial reduction in malaria cases after the final round of IRS relative to the baseline period until (adjusted IRR in the first month after IRS = 0.19, 95% CI 0.09–0.42) about 4 to 5 months after the final IRS campaign when malaria cases began to increase. Over the 10–31 months after IRS was stopped, the number of malaria cases increased by over fivefold relative to the baseline period (adjusted IRR = 5.24, 95% CI 3.67–7.50). This corresponds to predicted case counts of near zero immediately following final IRS campaign followed by an increase to about 1000 cases per month at each site (Fig. 3). These results were consistent when considering only laboratory-confirmed cases unadjusted for testing rates (Supplementary Fig. 3).

**Impact of restarting IRS with a single round**. A total of 858,380 outpatient visits were recorded across the analysis period for the nine sites (Table 2). Mean monthly malaria cases ranged from 643–1569 and the TPR ranged from 56.5–84.7% during the baseline period. These ranges were 501–762 and 48.5–72.0%, respectively during the evaluation period. Temporal trends of laboratory-confirmed malaria cases over time for the individual health facilities are presented in Supplementary Fig. 4.

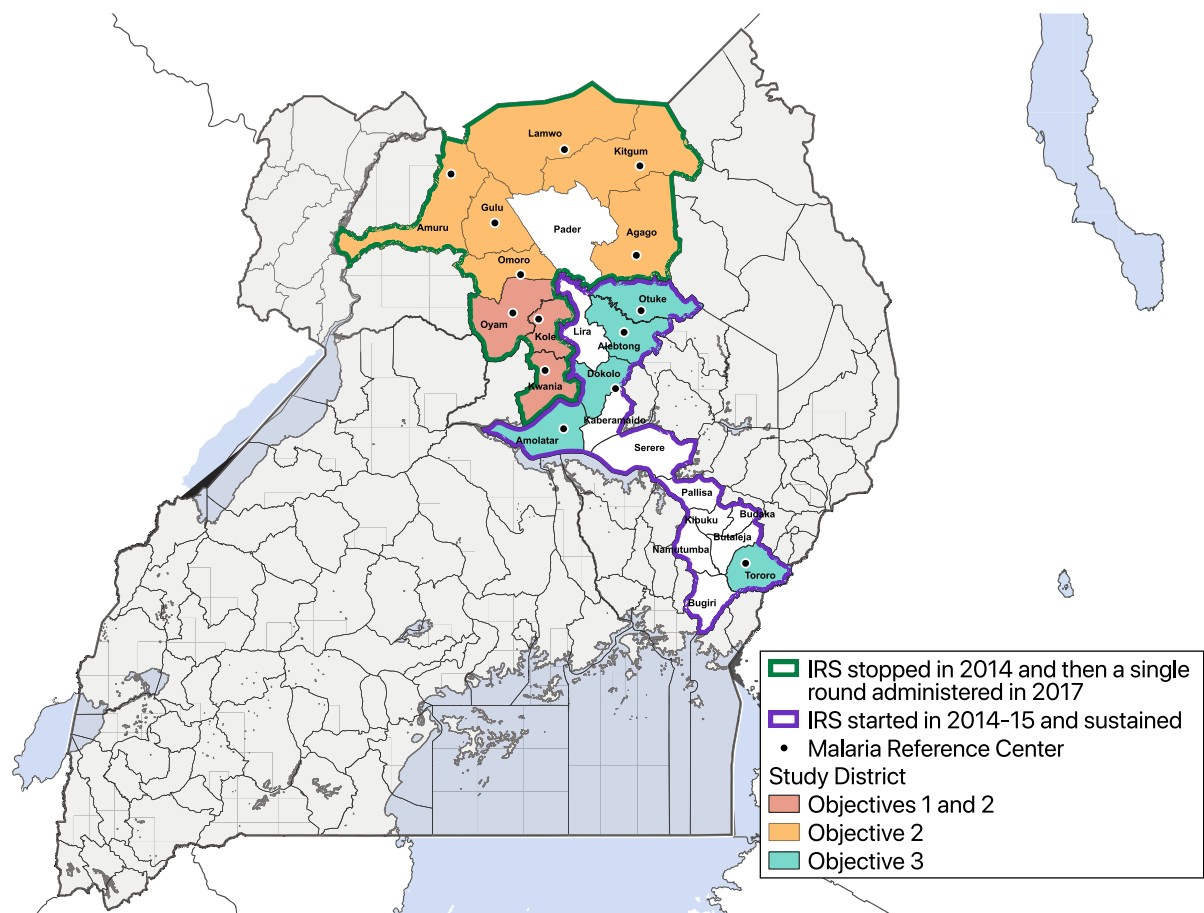

**Fig. 1 Map of Uganda showing study sites (Malaria Reference Centers) and indoor residual spraying (IRS) districts.** Districts not included in the analysis did not have an active Malaria Reference Center during the study period. Objective 1 is to assess the impact of withdrawing IRS after 5 years of sustained use; Objective 2 is to assess the impact of restarting IRS with a single round; and Objective 3 is to assess the impact of initiating and sustaining IRS.

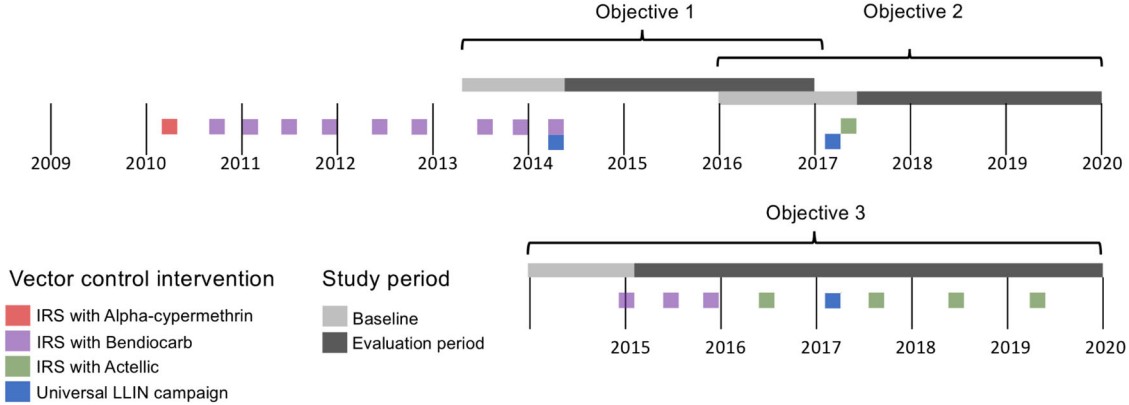

**Fig. 2 Timeline summarizing the dates of indoor residual spraying (IRS) campaigns, baseline, and evaluation periods.** Objective 1 is to assess the impact of withdrawing IRS after 5 years of sustained use; Objective 2 is to assess the impact of restarting IRS with a single round; and Objective 3 is to assess the impact of initiating and sustaining IRS. Exact dates of interventions are general; details on intervention dates by site are shown in Supplementary Fig. 1.

Monthly adjusted IRRs and 95% CI for the nine sites combined are presented in Fig. 4 and Supplementary Table 2. The single round of IRS was associated with a reduction in malaria cases until ~23 months post-IRS. Over the 8–12 months after the single round of IRS, malaria cases decreased by over fivefold relative to the baseline period (adjusted IRR = 0.17, 95% CI 0.15–0.20). After 23 months following the single round of IRS, malaria cases returned to a level similar to the baseline period before the single

round of IRS (adjusted IRR for months 23–31 = 1.06, 95% CI 0.92–1.21). These results were consistent when considering only laboratory-confirmed cases unadjusted for testing rates (Supplementary Fig. 5).

**Impact of initiating and sustaining IRS.** In total, 574,587 outpatient visits were observed across the five sites included in the analysis (Table 3). During the baseline period, average monthly

**Table 1 Summary statistics from health-facility based surveillance sites where IRS[a] was stopped after sustained use.**

| MRC[b] (District) | Time period | Number of months included | Total outpatient visits, n | Suspected malaria cases, n (% of total) | Tested for malaria, n (% of suspected) | RDT[c] performed (versus microscopy), n (% of tested) | Confirmed malaria cases, n (% of tested) | Confirmed cases adjusted for testing rate, n | Mean monthly confirmed cases adjusted for testing rate, n |
|---|---|---|---|---|---|---|---|---|---|
| Aboke HCIV (Kole) | Baseline | 9 | 14,015 | 3766 (26.9) | 3735 (99.2) | 2450 (65.6) | 923 (24.7) | 932 | 104 |
| | Evaluation | 25 | 46,850 | 21,245 (45.3) | 18,185 (85.6) | 17,210 (94.6) | 14,200 (78.0) | 16,699 | 668 |
| Aduku HCIV (Kwania) | Baseline | 12 | 23,899 | 13,425 (56.2) | 13,407 (99.9) | 955 (7.1) | 3189 (23.8) | 3193 | 266 |
| | Evaluation | 32 | 57,470 | 30,035 (52.2) | 25,896 (86.2) | 10,731 (41.4) | 13,537 (52.3) | 15,717 | 491 |
| Anyeke HCIV (Oyam) | Baseline | 8 | 15,859 | 3514 (22.2) | 2627 (74.8) | 2604 (99.1) | 680 (25.9) | 918 | 115 |
| | Evaluation | 25 | 66,501 | 28,755 (43.2) | 20,659 (71.8) | 16,147 (78.2) | 13,559 (65.6) | 18,774 | 751 |

[a]Indoor residual spraying.
[b]Malaria Reference Center.
[c]Rapid diagnostic test.

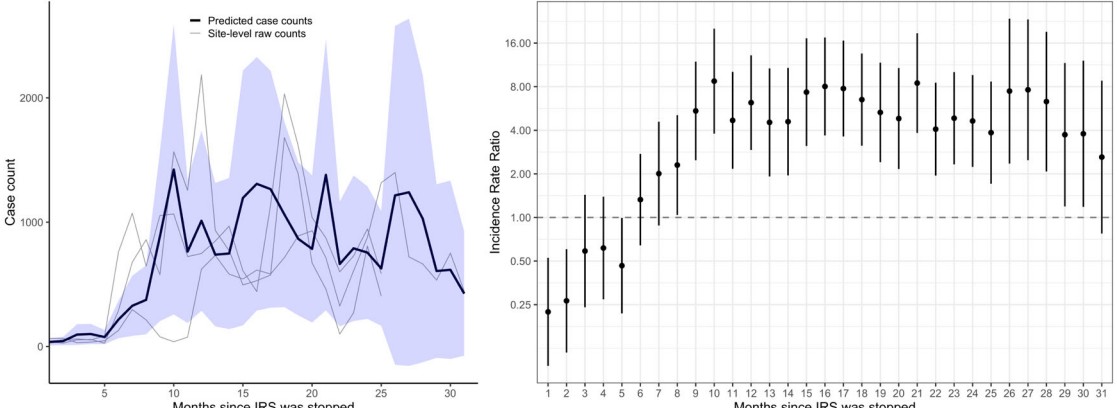

**Fig. 3 Predicted case counts and adjusted incidence rate ratios (IRR) from multilevel negative binomial model assessing the impact of withdrawing indoor residual spraying (IRS) after 5 years of sustained use.** The blue shaded region on the left represents the 95% confidence interval around the mean predicted case counts across sites from the adjusted regression model. Gray lines represent observed monthly case counts from individual sites. On the right, vertical bars represent the 95% confidence interval around the adjusted IRR (the measure of center for the error bars).

malaria cases adjusted for testing rates ranged from 286–657 and the TPR ranged from 25.4–67.0%. This range decreased to 85–289 for malaria cases and 13.8–45.3% for the TPR during the evaluation period. Temporal trends of laboratory-confirmed malaria cases over time for the individual health facilities are presented in Supplementary Fig. 6.

Monthly adjusted IRRs and 95% CI for the five sites combined are presented in Fig. 5 and Supplementary Table 3. There was a modest overall reduction in malaria case counts in the first 3 years after IRS was initiated relative to the baseline period, with some peaks in case counts returning to near baseline levels just prior to when rounds of IRS were administered. However, after the third year of sustained use, malaria case counts dropped substantially and remained low relative to the period before IRS was initiated. In the fourth and fifth year after IRS was initiated and sustained, malaria cases dropped by 85% (adjusted IRR = 0.15, 95% CI 0.12–0.18). These results were consistent when considering only laboratory-confirmed cases unadjusted for testing rates (Supplementary Fig. 7).

## Discussion

Uganda has been exceptionally successful in scaling-up coverage of LLINs. Following the mass distribution campaigns to deliver free LLINs in 2013–14 and 2017–18, 90 and 83% of households, respectively reported ownership of at least one LLIN[7,14]. However, despite this success, the burden of malaria remains high in much of the country. Uganda had the third highest number of malaria cases reported in 2019, with reported case incidence

increasing since 2014[2]. If Uganda is to achieve the goals established by the World Health Organization's Global Technical Strategy for malaria including reducing malaria case incidence by at least 90% by 2030 as compared with 2015[15], additional malaria control measures will be needed. This report highlights the critical role of IRS in substantially reducing the burden of malaria in areas where transmission remains high despite deployment of LLINs. Withdrawing IRS after 5 years of sustained use in three districts in northern Uganda was associated with a more than fivefold increase in malaria cases within 10 months. Restarting IRS with a single round in nine districts in Northern Uganda ~3 years after IRS had been stopped was associated with a transient but important (more than a fivefold) decrease in malaria cases within 8–12 months, returning to pre-IRS levels after 23 months. Initiating and sustaining IRS in five districts in Eastern Uganda was associated with a gradual reduction in malaria cases reaching almost a sevenfold reduction after 4–5 years.

Robust evidence supports the widespread use of LLINs for malaria control. In a systematic review of clinical trials conducted between 1987 and 2001, insecticide treated nets reduced all cause child mortality by 17% and the incidence of uncomplicated *P. falciparum* malaria by almost half[16]. However, there is concern that the effectiveness of LLINs may be diminishing due to widespread resistance to pyrethroids which until recently were the only class of insecticides approved for LLINs. Similar to many other African countries, high-level resistance to pyrethroids among the principle *Anopheles* vectors has been reported recently throughout Uganda[17–19]. In addition, behavioral changes in vector biting activity following the introduction of LLINs have been reported

**Table 2 Summary statistics from health-facility based surveillance sites that received a single round of IRS[a].**

| MRC[b] (District) | Time period | Number of months included | Total outpatient visits, n | Suspected malaria cases, n (% of total) | Tested for malaria, n (% of suspected) | RDT[c] performed (versus microscopy), n (% of tested) | Confirmed malaria cases, n (% of tested) | Confirmed cases adjusted for testing rate, n | Mean monthly confirmed cases adjusted for testing rate, n |
|---|---|---|---|---|---|---|---|---|---|
| Aboke HCIV (Kole) | Baseline | 12 | 18,361 | 10,247 (55.8) | 8161 (79.6) | 7663 (93.9) | 6069 (74.4) | 7740 | 645 |
| | Evaluation | 34 | 54,826 | 30,973 (56.5) | 30,674 (99.0) | 29,064 (94.8) | 22,097 (72.0) | 22,308 | 656 |
| Aduku HCIV (Kwania) | Baseline | 12 | 25,439 | 14,912 (58.6) | 11,944 (80.1) | 4854 (40.6) | 6559 (54.9) | 8009 | 667 |
| | Evaluation | 31 | 65,379 | 32,260 (49.3) | 31,337 (97.1) | 20,385 (65.1) | 15,201 (48.5) | 15,534 | 501 |
| Anyeke HCIV (Oyam) | Baseline | 12 | 30,447 | 15,873 (52.1) | 11,324 (71.3) | 8628 (76.2) | 7947 (70.2) | 11,018 | 918 |
| | Evaluation | 34 | 70,149 | 33,618 (47.9) | 32,522 (96.7) | 31,208 (96.0) | 21,799 (67.0) | 22,375 | 658 |
| Awach HCIV (Gulu) | Baseline | 12 | 27,375 | 16,788 (61.3) | 15,124 (90.1) | 14,932 (98.7) | 11,293 (74.7) | 12,558 | 1047 |
| | Evaluation | 30 | 69,375 | 36,760 (53.0) | 35,189 (95.7) | 34,070 (96.8) | 21,879 (62.2) | 22,851 | 762 |
| Lalogi HCIV (Omoro) | Baseline | 12 | 39,517 | 24,235 (61.3) | 23,959 (98.9) | 23,951 (99.9) | 17,000 (71.0) | 17,202 | 1,434 |
| | Evaluation | 31 | 72,449 | 41,846 (57.8) | 41,668 (99.6) | 40,804 (97.9) | 22,986 (55.2) | 23,060 | 744 |
| Patongo HCIII (Agago) | Baseline | 12 | 21,745 | 13,482 (62.0) | 13,244 (98.2) | 12,938 (97.7) | 10,032 (75.7) | 10,142 | 845 |
| | Evaluation | 34 | 54,486 | 34,482 (63.3) | 33,797 (98.0) | 32,176 (95.2) | 17,231 (51.0) | 17,440 | 513 |
| Atiak HCIV (Amuru) | Baseline | 12 | 33,077 | 22,250 (67.3) | 19,224 (86.4) | 19,151 (99.6) | 16,450 (85.6) | 19,044 | 1,587 |
| | Evaluation | 34 | 60,750 | 31,650 (52.1) | 30,754 (97.2) | 30,541 (99.3) | 19,766 (64.3) | 20,325 | 598 |
| Padibe HCIV (Lamwo) | Baseline | 12 | 15,967 | 10,212 (64.0) | 10,096 (98.9) | 10,089 (99.9) | 8171 (80.9) | 10,089 | 841 |
| | Evaluation | 28 | 50,117 | 26,883 (53.6) | 26,831 (99.8) | 25,956 (96.7) | 15,199 (56.6) | 15,224 | 544 |
| Namokora HCIV (Kitgum) | Baseline | 12 | 20,291 | 16,969 (83.6) | 15,991 (94.2) | 14,918 (93.3) | 10,049 (62.8) | 10,722 | 894 |
| | Evaluation | 31 | 56,765 | 40,185 (70.8) | 39,966 (99.5) | 38,468 (96.3) | 21,958 (54.9) | 22,063 | 712 |

[a]Indoor residual spraying.
[b]Malaria Reference Center.
[c]Rapid diagnostic test.

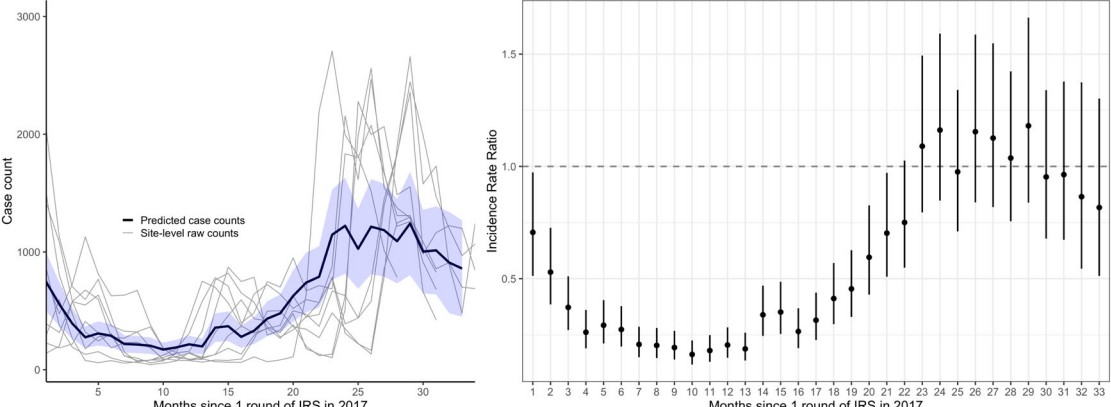

**Fig. 4 Predicted case counts and adjusted incidence rate ratios (IRR) from multilevel negative binomial model assessing the impact of restarting indoor residual spraying (IRS) with a single round.** The blue shaded region on the left represents the 95% confidence interval around the mean predicted case counts across sites from the adjusted regression model. Gray lines represent observed monthly case counts from individual sites. On the right, vertical bars represent the 95% confidence interval around the adjusted IRR (the measure of center for the error bars).

which could present new challenges for malaria control[20–22]. Finally, the effectiveness of LLINs may be further compromised by poor adherence and waning coverage in the setting of free distribution campaigns done intermittently. In Uganda, less than 18% of households reported adequate coverage (defined as at least one LLIN per two residents) 3 years after the 2013–14 distribution campaign[23] and adequate coverage decreased from 71% to 51% between 6 and 18 months following the 2017–18 distribution campaign[24]. Although the World Health Organization recommends mass distribution campaigns every 3 years, mounting evidence suggests that LLINs should be distributed more frequently to sustain high coverage[25–31].

Given concerns about the current effectiveness of pyrethroid-based LLINs and the persistently high burden of malaria despite aggressive scale-up of LLINs in countries like Uganda, additional malaria control measures are needed. IRS is an attractive option. Historically, IRS programs were used to dramatically reduce and even eliminate malaria in many parts of the world. Thus, while there is some evidence for the impact of IRS in the absence of LLINs[32], it is surprising that the evidence base from contemporary controlled trials on the impact of adding IRS to

LLINs for vector control is limited. A recent systematic review of cluster randomized controlled trials conducted in sub-Saharan Africa since 2008, reported that adding IRS using a "pyrethroid-like" insecticide to LLINs did not provide any benefits, while adding IRS with a "non-pyrethroid-like" insecticide produced mixed results[5]. Among the four trials comparing IRS plus LLINs with LLINs alone, three evaluated IRS with a carbamate (bendiocarb) and one evaluated a long-lasting organophosphate, pirimiphos-methyl (Actellic 300CS®)[33–36]. Only two trials (both using bendiocarb) assessed malaria incidence; one from Sudan found a 35% reduction when adding IRS to LLINs[34], while another from Benin found no benefit of adding IRS[33]. All four trials assessed parasite prevalence, with an overall non-significant trend towards a lower prevalence when adding IRS to LLINs (RR = 0.67, 95% CI 0.35–1.28)[5]. However, when the analyses were restricted to include only the two studies with LLIN usage over 50%, adding IRS reduced parasite prevalence by over 50% (RR = 0.47, 95% CI 0.33–0.67)[5]. Of note, none of the trials that evaluated the impact of adding IRS with a "non-pyrethroid-like" insecticide assessed outcomes beyond 2 years. More recently, a number of observational studies have reported

**Table 3 Summary statistics from health-facility based surveillance sites where IRS[a] was initiated and sustained.**

| MRC[b] (District) | Time period | Number of months included | Total outpatient visits, n | Suspected malaria cases, n (%) | Tested for malaria, n (%) | RDT[c] performed (versus microscopy), n (% of tested) | Confirmed malaria cases, n (%) | Confirmed malaria cases adjusted for testing rate, n | Mean monthly confirmed cases adjusted for testing rate, n |
|---|---|---|---|---|---|---|---|---|---|
| Nagongera HCIV (Tororo) | Baseline | 12 | 20,828 | 13,251 (63.6) | 13,096 (98.8) | 760 (5.8) | 3298 (25.2) | 3337 | 278 |
| | Evaluation | 59 | 97,012 | 36,308 (37.4) | 36,069 (99.3) | 13,129 (36.4) | 4984 (13.8) | 5022 | 85 |
| Amolatar HCIV (Amolatar) | Baseline | 12 | 19,552 | 8547 (43.7) | 6512 (76.2) | 5923 (91.0) | 3701 (56.8) | 4845 | 404 |
| | Evaluation | 59 | 89,779 | 24,889 (27.8) | 21,849 (87.9) | 19,459 (89.1) | 4822 (22.1) | 5854 | 99 |
| Dokolo HCIV (Dokolo) | Baseline | 12 | 25,570 | 12,854 (50.3) | 8875 (69.0) | 8212 (92.5) | 5211 (58.7) | 7889 | 657 |
| | Evaluation | 59 | 129,245 | 46,428 (35.9) | 44,972 (96.9) | 42,259 (94.0) | 10,210 (22.7) | 10,761 | 183 |
| Orum HCIV (Otuke) | Baseline | 11 | 16,120 | 9324 (57.8) | 8929 (95.8) | 3990 (44.7) | 5974 (66.9) | 6236 | 567 |
| | Evaluation | 59 | 65,036 | 37,430 (57.6) | 36,371 (97.2) | 19,536 (53.7) | 16,481 (45.3) | 17,069 | 289 |
| Alebtong HCIV (Alebtong) | Baseline | 8 | 15,359 | 6694 (43.6) | 4789 (71.5) | 4620 (96.5) | 3209 (67.0) | 4317 | 540 |
| | Evaluation | 59 | 94,055 | 40,821 (43.0) | 36,211 (88.7) | 32,327 (89.3) | 12,037 (33.2) | 13,869 | 235 |

[a]Indoor residual spraying.
[b]Malaria Reference Center.
[c]Rapid diagnostic test.

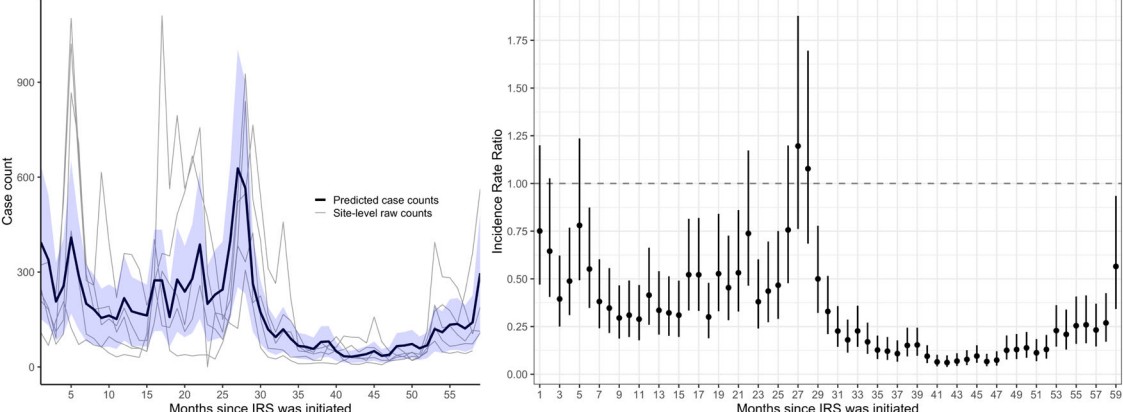

**Fig. 5 Predicted case counts and adjusted incidence rate ratios (IRR) from multilevel negative binomial model assessing the impact of initiating and sustaining indoor residual spraying (IRS).** The blue shaded region on the left represents the 95% confidence interval around the mean predicted case counts across sites from the adjusted regression model. Gray lines represent observed monthly case counts from individual sites. On the right, vertical bars represent the 95% confidence interval around the adjusted IRR (the measure of center for the error bars).

benefits of using IRS with pirimiphos-methyl (Actellic 300CS®). In the Mopti Region of Mali, delivery of a single round of IRS with Actellic 300CS® was associated with a 42% decrease in the peak incidence of laboratory-confirmed malaria cases reported at public health facilities[37]. In the Koulikoro Region of Mali, villages that received a single round of IRS with Actellic 300CS® combined with LLINs observed a greater than 50% decrease in the incidence of malaria compared to villages that only received LLINs[38]. In the Northern Region of Ghana, districts that received IRS with Actellic 300CS® reported 26–58% fewer cases of laboratory-confirmed malaria cases reported at public health facilities over a 2-year period, compared to districts that did not receive IRS[39]. In Northern Zambia, implementation of IRS with Actellic 300CS® targeting only high burden areas over a 3 year period was associated with a 25% decline in parasite prevalence during the rainy season, but no decline during the dry season[40]. In Western Kenya, the introduction of a single round of IRS with Actellic 300CS® was associated with a 44–65% decrease in district level malaria case counts over a 10 month period compared to pre-IRS levels[41]. In addition, several recent reports have documented dramatic resurgences of malaria following the withdrawal of IRS with bendiocarb in Benin[42], and the withdrawal of IRS with Actellic 300CS® in Mali and Ghana[37,39].

The results from this study provides additional support for the critical role IRS can play in reducing the burden of malaria in African countries with high LLINs coverage. A strength of the study was its use of a large, rigorously collected dataset. Data were

collected over nearly 7 years through an enhanced health facility-based surveillance system covering 14 districts in Uganda where IRS was being withdrawn, re-started, and initiated. This enhanced surveillance system facilitated laboratory testing and provided prospectively collected, individual-level data, allowing for analyses of quantitative changes in laboratory-confirmed cases of malaria over time, controlling for temporal changes in rainfall, seasonal effects, diagnostic practices, and health seeking behavior. Previous work by our group documented a marked decrease in malaria TPRs after 4 years of sustained IRS with bendiocarb in one district of Northern Uganda followed by a rapid resurgence over an 18-month period after IRS was withdrawn[11]. In this study we expand on these findings by including data from three districts and covering a 31-month period following the withdrawal of IRS. We were able to quantify more than a fivefold increase in malaria cases which was sustained over the 10–31 months following the withdrawal of IRS. This marked resurgence occurred despite the fact the first universal LLIN distribution campaign was timed to occur right after IRS was withdrawn. Given the dramatic nature of the resurgence, the Ugandan government was able to procure funding for a single round of IRS with Actellic 300CS® ~3 years after IRS was withdrawn in 10 districts of Northern Uganda. In this study, we assessed the impact of this single round in nine of these districts. This single round was associated with over a fivefold decrease in malaria cases after 8–12 months, with malaria cases returning to pre-IRS levels after almost 2 years. These data suggest that IRS with longer-acting formulations such as Actellic

300CS® administered every 2 years could be considered as a strategy for mitigating the risk of resurgence following sustained IRS and/or enabling countries to expand coverage when resources are limited, but formal assessment and a cost-effectiveness analyses are needed. This study also evaluated the impact of 5 years of sustained IRS in five districts of Eastern Uganda, starting first with bendiocarb and then switching to Actellic 300CS® after 18 months. Rounds of IRS were initially associated with marked decreases in malaria cases followed by peaks before subsequent rounds until the fourth and fifth years after IRS was initiated when there was a sustained decrease of almost sevenfold compared to pre-IRS level. Given the before-and-after nature of our study design, it is not clear whether the maximum sustained benefits of IRS seen after 4–5 years were due to the cumulative effect of multiple rounds of IRS, the switch from bendiocarb to Actellic 300CS®, improvements in implementation (although campaigns occurred regularly and coverage was universally high across rounds, see Supplementary Table 4), the second universal LLIN distribution campaign which occurred in this area in 2017, and/or other factors.

This study had several limitations. First, we used an observational study design, with measures of impact based on comparisons made before-and-after key changes in IRS policy. Although cluster randomized controlled trials are the gold standard study design for estimating the impact of IRS, it could be argued that withholding IRS would be unethical, given what is known about its impact in Uganda. Second, our estimates of impact could have been confounded by secular trends in factors not accounted for in our analyses. However, we feel that our overall conclusions are robust given the large amount of data available from multiple sites over an extended period with multiple complementary objectives providing consistent findings. Third, we could not assess the impact of IRS independent of LLIN use and did not have access to measures of IRS or LLIN coverage from our study populations. It is possible that some of the impacts we observed were from LLIN distributions in combination with IRS campaigns. However, we were able to provide a "real world" assessment of IRS in a setting where LLIN use is strongly supported by repeated universal distribution campaigns that are becoming increasingly common in sub-Saharan Africa. Similarly, we cannot draw conclusions on the impact of different IRS compounds given all sites received the same formulations consecutively. The results from Objective 3 indicate that malaria incidence dropped substantially in the years that districts stopped receiving bendiocarb and began receiving Actellic 300CS®. However, we cannot conclude whether this reduction was a result of this change or rather the cumulative impact of sustained IRS campaigns, as it has been suggested that in very high transmission settings, several years of IRS may be needed to maximize impact on measures of morbidity.[43,44] Finally, our study outcome was limited to case counts of laboratory-confirmed malaria captured at health facilities. Thus, we were unable to measure the impact of IRS on other important indicators such as measures of vector distribution, parasite prevalence, or mortality.

There is a growing body of evidence that combining LLINs with IRS using "non-pyrethroid-like" insecticides, especially the long acting organophosphate Actellic 300CS®, is highly effective at reducing the burden of malaria in Uganda, and elsewhere in Africa. Despite these encouraging findings, IRS coverage in Africa has been moving in the wrong direction. The proportion of those at risk protected by IRS in Africa peaked at just over 10% in 2010. However, the spread of pyrethroid resistance has led many control programs to switch to more expensive formulations resulting in a 53% decrease in the number of houses sprayed between years of peak coverage and 2015 across 18 countries supported by the US President's Malaria Initiative[45] and an overall reduction in the proportion protected by IRS in Africa to less than 2% in 2019[2]. Given the lack of recent progress in reducing the global burden of malaria coupled with challenges in funding, renewed commitments are needed to address the "high burden to high impact" approach now being advocated by the World Health Organization[2]. IRS is a widely available tool that could be scaled up, however demands currently exceed the availability of resources. Additional work is needed to optimize the use of IRS, prevent further spread of insecticide resistance, and better evaluate the cost-effectiveness of IRS in the context of other control interventions.

## Methods

**Health-facility based surveillance.** Enhanced malaria surveillance was established by UMSP in 2006[46]. UMSP operates Malaria Reference Centers (MRCs) at 70 level III/IV public health facilities across Uganda. At each MRC, individual-level data from standardized registers for all patients presenting to the outpatient departments are entered into a Microsoft Access (v16.0) database by on-site data entry officers. Variables include patient demographics, results of laboratory testing for malaria (rapid diagnostic test [RDT] or microscopy), diagnoses given, and treatments prescribed. Emphasis is placed on ensuring that patients with suspected malaria undergo testing, by either RDT or microscopy.

This study utilized data from 14 MRCs located in districts that either previously had IRS or have ongoing IRS campaigns. We estimated the impact of withdrawing IRS using data from three sites in Northern Uganda that had at least 6 months of data preceding the final round of IRS administered in 2014. To estimate the impact of restarting IRS with a single round administered in 2017, we used data from nine sites in Northern Uganda. To estimate the impact of sustained IRS over 5 years, we used data from five sites in Eastern Uganda where IRS had been implemented since 2014–15. For all analyses, we accessed raw, individual-level data, and aggregated to monthly-level data.

**Measures.** The exposure was specified as an indicator variable for each month since IRS was withdrawn or initiated relative to a baseline period (Fig. 1 and Supplementary Fig. 1). All baseline periods were defined as the 12 month period immediately preceding the intervention (or stopping the intervention) pending data availability. If fewer than 12 months of baseline data were available, we included the maximum amount of time available for sites that had at least 6 months of data before the evaluation period. We also fit separate models with categorical exposure variables divided into distinct periods of months. To determine the impact of withdrawing IRS after at least 5 years of sustained use, the baseline period was defined as the year leading up to the final round of IRS, and the evaluation period lasted through 2016, prior to when an additional round of IRS was implemented. In order to determine the impact of restarting IRS with a single round of IRS, the baseline period was defined as 1 year prior to the single round of IRS and the evaluation period went through December 2019. To determine the impact of initiating and sustaining IRS, the baseline period was the year prior to IRS initiation, and the evaluation period lasted through December 2019.

The primary outcome was the monthly count of laboratory-confirmed malaria cases at each MRC. The case count was corrected for testing rates by multiplying the number of individuals with suspected malaria but not tested each month by the TPR (the number who tested positive divided by the total number tested) for that month and adding the result to the number of laboratory-confirmed positive cases. As a sensitivity analysis, we re-specified the models including only laboratory-confirmed case counts as the outcome.

We adjusted for time-varying variables that impact malaria burden and malaria case detection at the health facility. These variables included monthly rainfall at the health facility lagged by 1 month extracted from the Climate Hazards Infrared Precipitation with Stations database[47], indicator variables for month of the year (to adjust for seasonal effects), the proportion of tests that were RDTs in that month (vs. microscopy), and the number of individuals who attended the health facility but were not suspected of having malaria in that month (to adjust for potential changes in care-seeking behaviors over time).

**Statistical analysis.** For each objective, we specified mixed effects negative binomial regression models with random intercepts for health facility. Coefficients for the exposure variable were exponentiated to represent the incidence rate ratio (IRR) comparing the incidence of malaria in the month of interest relative to the baseline period. This method assumes that the underlying population has remained constant over the study period. All analyses were carried out in R v3.6 and Stata v14.

**Ethics Approval and consent to participate.** Ethical approval for study procedures and data collection was provided by ethics committees of University of California San Francisco (REF 250046), the School of Medicine College of Health Sciences at Makerere University (REF 2019-087), and Uganda National Council of

Science and Technology (REF HS 2659). Written informed consent was not required by the ethical review committees due to the routine, de-identified nature of the data.

**Reporting Summary**. Further information on research design is available in the Nature Research Reporting Summary linked to this article.

## Data availability

The datasets generated during and/or analysed during the current study are available in github with the identifier https://doi.org/10.5281/zenodo.4625804. To obtain raw data for these analyses, please contact the corresponding author. Precipitation data was accessed through the Climate Hazards Infrared Precipitations with Stations database, available publicly at https://data.chc.ucsb.edu/products/CHIRPS-2.0/[48]

## Code availability

Computing code is available at https://doi.org/10.5281/zenodo.4625804

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

## Acknowledgements

We would like to acknowledge the health workers at all 14 health facilities that contributed data for this study. We would like to thank the Ugandan Ministry of Health National Malaria Control Division, and USAID—President's Malaria Initiative. This work was supported by the National Institutes of Health as part of the International Centers of Excellence in Malaria Research (ICMER) program (U19AI089674). A.E. is supported by the National Institute of Allergy and Infectious Diseases (F31AI150029). J.I. N. is supported by the Fogarty International Center (Emerging Global Leader Award grant number K43TW010365). E.A. is supported by the Fogarty International Center of the National Institutes of Health under Award Number D43TW010526.

## Author contributions

J.F.N., A.E., G.D., and I.R.-B. conceived of the study. J.F.N. led the data collection activities with support from J.I.N., A.M., M.K., A.S., J.K., E.A., S.G., C.E., S.G.S., C.M.-S., and M.R.K. A.E. and I.R.-B. led the data analysis with support from G.D. A.E. and J.F.N. drafted the manuscript with support from G.D., S.G.S., and I.R.-B. J.I.N., A.M., M.K., A.S., J.K., E.A., S.G., J.O., C.E., J.S., M.O., D.R., C.M.-S., K.B., and M.R.K. contributed to interpretation of the results and edited the manuscripts. All authors read and approved the final manuscript.

## Competing interests

The authors declare no competing interests.
