## [Peer Review File · Nature Communications]

REVIEWER COMMENTS

Reviewer #1 (Remarks to the Author):

Namuganga et al present estimates of the impact of stopping and starting IRS on malaria cases reported in health systems across 14 districts in northern Uganda. This is an important topic as IRS is increasingly being considered in high burden countries in Africa and there is a lack of evidence of the added benefit in areas with high LLIN coverage in real world settings. These data presented here appear strong and the statistical analyses is appropriate though the rationale could be better described at times. Overall, the manuscript is a useful and of high quality though there could be some minor improvement in the presentation.

Major Points

1. The authors do not state what type of LLINs are distributed. Some regions of Uganda received pyrethroid only LLINs in the last mass campaign whilst others received nets with the synergist PBO. As some co-authors have shown PBO LLINs have a bigger impact it seems pertinent to include this in the analyses here. If there were different LLINs allocated to the communities evaluated then it would be good to see if this made a statistical difference as this is an important question (i.e. it makes sense that in areas with pyrethroid resistance a bigger public health impact would be seen adding IRS to areas with standard pyrethroid nets). If not then it should be stated so as not to cause confusion.

2. I found the timing of the baseline confusing. The text says 1 year was used (which makes sense) but this does not match with Figure 5 which indicates a longer (and variable) baseline and Table 1 gives different values again. This is not helped by the x-axes on Figures 1-3 which could be improved. It is clear that timing of IRS varied between the districts so date cannot be used but showing the period of baseline data would help people interpret the IRR (i.e. case counts for the 12 months prior to 0). Finally, the baseline is defined in Objective 1 as "period was defined as the final year of sustained IRS use". Given people don't use IRS (it is done at a time point and then has a prolonged impact) this definition seems confusing. Was it 1 year after the last use of the intervention or 1 year after the intervention had worn off?

3. Not all IRS is equal. Nevertheless the analysis groups all types together and differences between products are not discussed in the results. This may create confusion for Figure 3. Currently the text gave the impression that IRS started working better after 3 years without giving an explanation. It appears likely this was driven from the switch from bendiocarb to actellic from Figure S5. Could this not be investigated in the regression? Also, the impact of the next round of mass bednet distribution could be included as a variable if this information were available and would be really informative. The discussion says this wouldn't be possible in the regression but was it tried (there seems to be some of the variability between sites)? If this is really not the case then it should be flagged more prominently in the discussion, particular in relation to other results as it is also shown in objective 1 (where it bounces back after bendiocarb after 10 months) but the transient effect of Actellic lasts 23 months in objective 2.

4. It says that covariates such as rainfall, test type, health care attendance were adjusted for but more description of how this was done or whether it made an impact would help understanding.

Minor points.

5. Paper ordering. The ordering of the figures could be changed to help the reader. Figures 4 and 5 are good, but would be more informative at the beginning of the results section (especially 5) to facilitate comprehension of the results. Suggest reordering.

6. Figures 1-3 are good though the caption could provide more detail. The legend for the grey line refers to "site", is this not the same as district?

7. Line 60 (and something similar on 183) says that "until recently pyrethroids were the only class of insecticide approved for use in LLINs". Approved by whom and what do you mean of by approval? This is important because non-pyrethroid nets are not recommended by the WHO as

there is no evidence that non-pyrethroid insecticides have public health benefits on LLINs.

8. Line 95 "TPR" should be defined explicitly.

9. Line 211 says that adding IRS was only significant when the was >50% LLIN coverage. Is this true or a typo, I haven't gone back to the original reference but I would have thought it was <50% when things became significant. Please excuse my laziness if this is the case.

10. Line 234 "which is a strength of study" seems a clumsy sentence, suggest revising.

11. The study assumes a constant population size. Is this fair for the region over 6 years?

12. Could LLIN mass distribution be added to Fig S5?

Reviewer #2 (Remarks to the Author):

The authors evaluated the impact of IRS on malaria in Uganda using data from a large network of health facility-based malaria surveillance sites. The study, which relies on complex design, aims to follow up malaria incidence in 14 different districts where IRS campaigns were either stopped, stopped and reintroduced or sustained. This "observational" study (before-after) showed that sustaining IRS has drastically reduced malaria incidence by >80% while stopping IRS led to a 5 fold increase in the number of malaria cases within a year. All together these findings highlight that sustaining core vector control intervention such as long lasting IRS is key to achieving malaria goals in the country. Although we know that results from observational studies can be influenced by unpredictable confounding factors (a weakness of the study), the statistical method used to analyse the large datasets was adequate and provides high confidence in the results. Overall the results are well presented, the paper is well written and the article is suitable for publication in Nature Communications after revisions.

Main comments

The study deals with complex study design due to the discontinued use of IRS campaigns in areas where the study took place. IRS treatments were sporadically done in areas where IRS has been considered as interrupted for many years which make interpretation of the results difficult (considering the proximity of the different districts and the time-lag needed to observe an effect). I thank the authors however for providing clear chart (fig 5) summarizing the dates of IRS treatments including baseline and evaluation periods.

In the Objective 3, the authors showed 85% drop in malaria prevalence 4 to 5 years after initiating and sustaining IRS (in the 5 districts located in the South). I wonder why it took so long time to get such "outstanding" results since no reduction was seen during the first 3 years (this period of time is normally sufficient to observe an effect when the treatment is properly done)? Does the authors reported any problem with the implementation of IRS (inappropriate dose, frequency of application, coverage, ?). I think they should provide more information on IRS treatment as this may strongly impact on the outcomes (eg Targeted dose, frequency of application, coverage per district, etc). A specific table could be provided in the paper. By experience, I have doubts also about the fact that IRS alone was responsible for such drop in malaria cases in a single year (between Y3 to Y4 post intervention); Are the authors aware about any other actions (drugs, LLIN, larviciding, etc) introduced in the study area at this period that could have boosted the IRS effect?

I also wonder why MRCs data from 9 districts of the study area (1 district in the north "Pader" and 8 in the South) were not included in the analysis ("white districts in fig 4)? Did they not meet criteria of selection or did they receive different vector control treatments than the one's included in the analysis? This is important to know considering the complex study design.

P18 Did the authors adjusted for other variables that may impact on malaria infections (age, sex, antimalarial treatment, IRS coverage, LLIN use, etc) ?

L93. Please replace "withdrawing" by "stopping" or "interrupting" IRS as the intention of the MoH was probably not to "remove" an effective treatment from the population (it sounds that we remove something belonging to the community here...).

L198. Overall the evidence supporting the effectiveness of IRS in sub sahara Africa (either alone or in combination with LLIN) is rather limited... The authors forgot to talk about the Cochrane review of Pluess et al 2010 that showed limited evidence for supporting IRS for malaria control in stable malaria settings (Cochrane Review published in 2010). Please add a statement in the text.

L253. The authors should be more careful with the following statement "longer-acting formulations such as Actellic 300CS® administered every 2 years may be a cost-effective strategy for mitigating the risk of resurgence following sustained IRS and/or enabling countries to expand coverage when resources are limited).

First there's no data on the residual activity of Actellic 300CS IRS in the paper so we don't know for how long the treatment has remained effective in the study's conditions (eg no residual bioassays on the walls were conducted by the team over the course of the study). Since the study was "observational (before-after and not a RCT) we cannot exclude the impact of other confounding factors on the outcomes (as highlighted by the authors in L261-263). Furthermore, there was no cost-assessment data provided in the paper so we can't conclude on the cost effectiveness of IRS compared to other VC interventions (eg what is the cost for a single round of Actellic 300 per year in the study area?). Actually the higher cost of new formulated products compared to conventional ones can lead to a reduction of IRS coverage and then increase of disease incidence. The authors have to revise their statement considering the lack of data to support their claim.

Minor comments

Title: Can the authors replace "starting" by "re introducing" or "re starting" as IRS were previously implemented in all study sites.

Abstract. L46 Please precise the endpoints used to evaluate the impact of IRS.

Introduction;

L55. Update data from the WHO malaria report 2020

L62. New LLIN using synergist (Olyset Plus, Permanet 3.0) or non-pyrethroids (G2 combining alphacypermethrin and chlorfenapyr) have been developed and distributed in Africa to improve malaria control in pyrethroid resistance area and the authors should reflect that in the text.

L76. Please indicate the insecticide and targeted dose used in IRS campaigns.

L228: resurgence "of malaria"

REVIEWER COMMENTS

Reviewer #1 (Remarks to the Author):

Namuganga et al present estimates of the impact of stopping and starting IRS on malaria cases reported in health systems across 14 districts in northern Uganda. This is an important topic as IRS is increasingly being considered in high burden countries in Africa and there is a lack of evidence of the added benefit in areas with high LLIN coverage in real world settings. These data presented here appear strong and the statistical analyses is appropriate though the rationale could be better described at times. Overall, the manuscript is a useful and of high quality though there could be some minor improvement in the presentation.

Major Points

1. The authors do not state what type of LLINs are distributed. Some regions of Uganda received pyrethroid only LLINs in the last mass campaign whilst others received nets with the synergist PBO. As some co-authors have shown PBO LLINs have a bigger impact it seems pertinent to include this in the analyses here. If there were different LLINs allocated to the communities evaluated then it would be good to see if this made a statistical difference as this is an important question (i.e. it makes sense that in areas with pyrethroid resistance a bigger public health impact would be seen adding IRS to areas with standard pyrethroid nets). If not then it should be stated so as not to cause confusion.

Thank you for noting this. In Uganda, there was no overlap between districts that received IRS and those that received PBO nets. In the Results, we have clarified that study districts received standard pyrethroid nets at both universal LLIN distributions: “Universal LLIN distribution campaigns were conducted in 2013-14 and 2017-18, where LLINs were distributed free-of-charge by the Uganda Ministry of Health targeting 1 LLIN for every two household residents. *In 2013-14, all districts across the country received LLINs impregnated with pyrethroid insecticides. In 2017-18, the Ministry of Health distributed both conventional LLINs and LLINs containing piperonyl butoxide (PBO) due to concerns of pyrethroid resistance. During the latter distribution, all districts included in this analysis received conventional pyrethroid insecticides due to prior concerns of antagonism between PBO LLINs and Actellic 300CS®.*”¹³

2. I found the timing of the baseline confusing. The text says 1 year was used (which makes sense) but this does not match with Figure 5 which indicates a longer (and variable) baseline and Table 1 gives different values again. This is not helped by the x-axes on Figures 1-3 which could be improved. It is clear that timing of IRS varied between the districts so date cannot be used but showing the period of baseline data would help people interpret the IRR (i.e. case counts for the 12 months prior to 0). Finally, the baseline is defined in Objective 1 as “period was defined as the final year of sustained IRS use”. Given people don’t use IRS (it is done at a time point and then has a prolonged impact) this definition seems confusing. Was it 1 year after the last use of the intervention or 1 year after the intervention had worn off?

We appreciate this point and have made some changes to the text and to the analysis for clarity. In the Methods, we have added (new text in Italics):

“The exposure was specified as an indicator variable for each month since IRS was withdrawn or initiated relative to a baseline period (Fig 1 and Supplementary Fig S1). All baseline periods were defined as the 12 month period immediately preceding the intervention (or stopping the intervention) pending data availability. If fewer than 12 months of baseline data were available, we included the maximum amount of time available for sites that had at least 6 months of data before the evaluation period.

For each of our objectives, the baseline periods were defined as follows. To determine the impact of withdrawing IRS after at least five years of sustained use, the baseline period was defined as the year leading up to the final round of IRS, and the evaluation period lasted through 2016, prior to when an additional round of IRS was implemented. In order to determine the impact of restarting IRS with a single round of IRS, the baseline period was defined as one year prior to the single round of IRS and the evaluation period went through December 2019. To determine the impact of initiating and sustaining IRS, the baseline period was the year prior to IRS initiation, and the evaluation period lasted through December 2019.”

This is now reflected in the analysis; all baseline periods are 12 months except for sites where this amount of time was not available. In these instances, we required at least 6 months of data to include the site in the analysis. Given baseline periods are not shown in Figures 3-5 (formerly Figures 1-3), we did not modify the axis labels.

3. Not all IRS is equal. Nevertheless the analysis groups all types together and differences between products are not discussed in the results. This may create confusion for Figure 3. Currently the text gave the impression that IRS started working better after 3 years without giving an explanation. It appears likely this was driven from the switch from bendiocarb to actellic from Figure S5. Could this not be investigated in the regression? Also, the impact of the next round of mass bednet distribution could be included as a variable if this information were available and would be really informative. The discussion says this wouldn't be possible in the regression but was it tried (there seems to be some of the variability between sites)? If this is really not the case then it should be flagged more prominently in the discussion, particular in relation to other results as it is also shown in objective 1 (where it bounces back after bendiocarb after 10 months) but the transient effect of Actellic lasts 23 months in objective 2.

These are valid points. Indeed, sites that began receiving IRS in 2014 first received an average of 3-4 rounds of Bendiocarb every 6 months followed by 2-3 rounds of Actellic annually. While it is true that the marked reduction in cases observed in the 4th and 5th year of sustained use coincides with the switch to Actellic, these data do not allow for us to draw conclusions regarding the efficacy of Bendiocarb versus Actellic. This is because all sites received one following the other in roughly the same time period. We have added this as a limitation in the Discussion: “Similarly, we cannot draw conclusions on the impact of different IRS compounds given all sites received the same formulations consecutively. The results from Objective 3 indicate that malaria incidence dropped substantially in the years that districts stopped receiving Bendiocarb and began receiving Actellic 300CS®. However, we cannot conclude whether this reduction was a result of this change or rather the cumulative impact of sustained IRS campaigns. It has been suggested that in very high transmission

settings, several years of IRS may be needed to maximize impact on measures of morbidity.^{43,44} See two references cited below this response on this latter point.

Similarly, given the fact that this reduction coincided with a distribution of LLINs, we cannot rule out their impact on the massive reduction of malaria incidence observed in the 4th and 5th year. Given all sites received LLINs at the same time, and IRS campaigns were occurring simultaneously, we are unable to generate an effect estimate for the impact of the LLIN campaign on its own. We do note in the discussion that a LLIN distribution accompanied the withdrawal of IRS for the three sites included in Objective 1 and, despite this, a large uptick in incidence was observed. We have added the LLIN distribution to all figures showing raw data (Fig S2, S4, and S6) so the reader can observe the timing of these distributions relative to IRS campaigns.

Citations:

Curtis CF, Mnzava AE. Comparison of house spraying and insecticide-treated nets for malaria control. *Bull World Health Organ.* 2000;78(12):1389-400. Epub 2003 Nov 17. PMID: 11196486; PMCID: PMC2560652.

Sharp BL, Kleinschmidt I, Streat E, Maharaj R, Barnes KI, Durrheim DN, Ridl FC, Morris N, Seocharan I, Kunene S, LA Grange JJ, Mthembu JD, Maartens F, Martin CL, Barreto A. Seven years of regional malaria control collaboration--Mozambique, South Africa, and Swaziland. *Am J Trop Med Hyg.* 2007 Jan;76(1):42-7. PMID: 17255227; PMCID: PMC3749812.

4. It says that covariates such as rainfall, test type, health care attendance were adjusted for but more description of how this was done or whether it made an impact would help understanding.

Thank you for this point. We have added more description of the covariates in the text. In addition, the Supplemental Tables with effect estimates from the regression models now include the coefficients for these covariates.

Minor points.

5. Paper ordering. The ordering of the figures could be changed to help the reader. Figures 4 and 5 are good, but would be more informative at the beginning of the results section (especially 5) to facilitate comprehension of the results. Suggest reordering.

We agree with this. In response, we have moved parts of the Methods (describing the study sites and intervention periods) to the Results. We have also changed the order of figures as suggested.

6. Figures 1-3 are good though the caption could provide more detail. The legend for the grey line refers to "site", is this not the same as district?

We have added more explanation to the figure legends for Figs 3-5 (formally Figs 1-3). Indeed, the grey lines refer to sites – these are monthly counts of cases at the health facility

(site) level. We hope the new text clarifies this (new text in Italics): “Adjusted IRR and predicted case counts from multilevel negative binomial model assessing the impact of withdrawing IRS after 5 years of sustained use. The blue shaded region represents the 95% confidence interval around the *mean* predicted case counts *across sites* from the adjusted regression model. *Grey lines represent observed monthly case counts from individual sites.* Vertical bars represent the 95% confidence interval around adjusted IRR.”

7. Line 60 (and something similar on 183) says that “until recently pyrethroids were the only class of insecticide approved for use in LLINs”. Approved by whom and what do you mean of by approval? This is important because non-pyrethroid nets are not recommended by the WHO as there is no evidence that non-pyrethroid insecticides have public health benefits on LLINs.

In response to this comment we have edited this sentence to read as follows: “Pyrethroids are the only class of insecticides widely use in LLINs and, given the emergence of pyrethroid resistance^{4,5}, there is concern that the effectiveness of LLINs may be diminishing.....”

8. Line 95 “TPR” should be defined explicitly.

We have added a definition of TPR in the text: “During the baseline period, average monthly cases ranged from 104-272 and test positivity rate (TPR), the proportion of individuals tested for malaria that resulted in a positive test, ranged from 23.7%-25.9%.”

9. Line 211 says that adding IRS was only significant when the was >50% LLIN coverage. Is this true or a typo, I haven’t gone back to the original reference but I would have thought it was <50% when things became significant. Please excuse my laziness if this is the case.

Indeed, this Cochrane Review suggests that LLINs+IRS is favored over LLINs alone when coverage of LLINs was greater than 50% (see Analysis 1.3 in Choi et al 2019).

10. Line 234 “which is a strength of study” seems a clumsy sentence, suggest revising.

We have edited this sentence: “A strength of the study was its use of a large, rigorously collected dataset.”

11. The study assumes a constant population size. Is this fair for the region over 6 years?

This is a fair point given that we do not include a population denominator over time. However, we do adjust for the monthly count of patients not suspected of having malaria who attended the facility for any reason, which should account at least in part for population growth. Below we plot the monthly count of these patients; there is no obvious

upward trend over time.

12. Could LLIN mass distribution be added to Fig S5?

We have added the LLIN distribution to this figure (now Fig S6) and to all supplemental figures showing raw data.

Reviewer #2 (Remarks to the Author):

The authors evaluated the impact of IRS on malaria in Uganda using data from a large network of health facility-based malaria surveillance sites. The study, which relies on complex design, aims to follow up malaria incidence in 14 different districts where IRS campaigns were either stopped, stopped and reintroduced or sustained. This “observational “study (before-after) showed that sustaining IRS has drastically reduced malaria incidence by >80% while stopping IRS led to a 5 fold increase in the number of malaria cases within a year. All together these findings highlight that sustaining core vector control intervention such as long lasting IRS is key to achieving malaria goals in the country. Although we know that results from observational studies can be influenced by unpredictable confounding factors (a weakness of the study), the statistical method used to analyse the large datasets was adequate and provides high confidence in the results. Overall the results are well presented, the paper is well written and the article is suitable for publication in Nature Communications after revisions.

Main comments

The study deals with complex study design due to the discontinued use of IRS campaigns in areas where the study took place. IRS treatments were sporadically done in areas where IRS has been considered as interrupted for many years which make interpretation of the results difficult (considering the proximity of the different districts and the time-lag needed to observe an effect). I thank the authors however for providing clear chart (fig 5) summarizing the dates of IRS treatments including baseline and evaluation periods.

In the Objective 3, the authors showed 85% drop in malaria prevalence 4 to 5 years after initiating and sustaining IRS (in the 5 districts located in the South). I wonder why it took so long time to get such “outstanding” results since no reduction was seen during the first 3 years (this period of time is normally sufficient to observe an effect when the treatment is properly done)? Does the authors reported any problem with the implementation of IRS (inappropriate dose, frequency of application, coverage, ?). I think they should provide more information on IRS treatment as this may strongly impact on the outcomes (eg Targeted dose, frequency of application, coverage per district, etc). A specific table could be provided in the paper. By experience, I have doubts also about the fact that IRS alone was responsible for such drop in malaria cases in a single year (between Y3 to Y4 post intervention); Are the authors aware about any other actions (drugs, LLIN, larviciding, etc) introduced in the study area at this period that could have boosted the IRS effect?

Thank you for noting this. In addition to the response to Reviewer 1’s comment #3, we include for your reference a table of IRS coverage per district during the rounds of Bendiocarb and Actellic (also included as Supplementary Table S4). Note that coverage is quite high throughout the rounds of spraying for which coverage data are available and the timing of spraying is not irregular. We therefore do not posit that the change in malaria incidence was due to implementation challenges in the first few years of IRS implementation. We include the following text in the Discussion (new text in Italics): “Given the before-and-after nature of our study design, it is not clear whether the maximum sustained benefits of IRS seen after 4-5 years were due to the cumulative effect of multiple rounds of IRS, the switch from bendiocarb to Actellic 300CS®, *improvements in implementation (although campaigns occurred regularly and coverage was universally high across rounds, see Supplementary Table S4)*, the second universal LLIN distribution campaign which occurred in this area in 2017, and/or other factors.”

It is important to underscore that while the largest reduction was observed during and after the third year, there were important reductions during the first two years as well (IRR between 0.5 and 0.75).

As we discussed in our comments to Reviewer 1, one explanation may be the necessity of several years of IRS to maximize impact in high transmission settings. In addition, we cannot rule out the potential impact of the LLIN distribution in 2017 in contributing to this reduction. We have emphasized our inability to tease out effects of LLINs and IRS as a limitation of our study (new text in Italics): “Third, we could not assess the impact of IRS independent of LLIN use and did not have access to measures of IRS or LLIN coverage from our study populations. *We cannot rule out that the impacts we observed were from LLIN distributions in combination with IRS campaigns.* However, we were able to provide a “real world” assessment of IRS in a setting where LLIN use is strongly supported by repeated universal distribution campaigns that are becoming increasingly common in sub-Saharan Africa.”

MRC	District		1 st round	2 nd round	3 rd round	4 th round	5 th round	6 th round	7 th round
Nagongera HCIV	Tororo	Insecticide	Bendiocarb	Bendiocarb	Bendiocarb	Actellic	Actellic	Actellic	Actellic
		Date	08-Dec-14 to 19-Feb-15	08-Jun-15 to 12-Jul-15	02-Nov-15 to 12-Dec-15	12-Jun-16 to 9-Jul-16	17-July-17 to 19-Aug-17	11-Jun-18 to 27-7-18	18-Mar-19 to 15-Apr-19
		Coverage	N/A	94.6%	96.3%	93%	94.9%	95.8%	92.1%
Amolatar HCIV	Amolatar	Insecticide	Bendiocarb	Bendiocarb	Bendiocarb	Bendiocarb	Actellic	Actellic	Actellic
		Date	08-Dec-14 to 19-Feb-15	08-Jun-15 to 12-Jul-15	12-Oct-15 to 07-Nov-15	24-Oct-16 to 19-Nov-16	2-May-17 to 6-June-17	9-Apr-18 to 12-May-18	27-May-19 to 27-June-19
		Coverage	N/A	96.1%	91.4%	94.1%	95.8%	96.1%	94.2%
Dokolo HCIV	Dokolo	Insecticide	Bendiocarb	Bendiocarb	Bendiocarb	Actellic	Actellic	Actellic	Sumishield 50W
		Date	08-Dec-14 to 19-Feb-15	08-Jun-15 to 12-Jul-15	12-Oct-15 to 07-Nov-15	18-Apr-16 to 9-Jul-16	2-May-17 to 6-June-17	9-Apr-18 to 12-May-18	27-May-19 to 27-June-19
		Coverage	N/A	88.7%	92.1%	94.8%	97.6%	94.3%	95.2%
Orum HCIV	Otuke	Insecticide	Bendiocarb	Bendiocarb	Bendiocarb	Actellic	Actellic	Actellic	Actellic
		Date	08-Dec-14 to 19-Feb-15	08-Jun-15 to 12-Jul-15	12-Oct-15 to 07-Nov-15	18-Apr-16 to 9-Jul-16	17-July-17 to 19-Aug-17	11-Jun-18 to 27-7-18	27-May-19 to 27-June-19
		Coverage	N/A	96.6%	94.8%	97.6%	96.7%	98.4%	98.8%
Alebtong HCIV	Alebtong	Insecticide	Bendiocarb	Bendiocarb	Bendiocarb	Bendiocarb	Actellic	Actellic	Actellic
		Date	08-Dec-14 to 19-Feb-15	08-Jun-15 to 12-Jul-15	12-Oct-15 to 07-Nov-15	24-Oct-16 to 19-Nov-16	2-May-17 to 6-June-17	9-Apr-18 to 12-May-18	27-May-19 to 27-June-19
		Coverage	N/A	96.9%	95.4%	97.5%	98.3%	90.9%	95.0%

Coverage data not available for the first IRS campaign in 2014.

I also wonder why MRCs data from 9 districts of the study area (1 district in the north “Pader” and 8 in the South) were not included in the analysis (“white districts in fig 4)? Did they not meet criteria of selection or did they receive different vector control treatments than the one’s included in the analysis? This is important to know considering the complex study design.

Districts that were not included in the analysis did not have an active Malaria Reference Center at the time of the study. This is now clarified in the Figure 4 legend.

P18 Did the authors adjusted for other variables that may impact on malaria infections (age, sex, antimalarial treatment, IRS coverage, LLIN use, etc) ?

Given this analysis was aggregated to case counts by month, we could not adjust for individual-level variables such as age, sex, and treatment.

We chose not to adjust for IRS coverage as we are interested in the impact of IRS campaigns over time, regardless of their coverage, not in estimating what the impact would have been if coverage were set to a specific value (such as 100%). In addition, coverage was universally high across rounds (see Supplementary Table S4), so we do not likely have adequate variation to determine the impact of IRS if coverage were lower. In addition, we unfortunately do not have data on LLIN use over time at these sites, so we could not include it as an adjustment.

L93. Please replace “withdrawing” by “stopping” or “interrupting” IRS as the intention of the

MoH was probably not to “remove” a effective treatment from the population (it sounds that we remove something belonging to the community here...).

The term “withdrawal” was used in this context by the Ugandan Ministry of Health National Malaria Control Division in their 2017-2018 annual report: “A universal net coverage campaign was conducted in Uganda and one round of IRS conducted in the 11 Northern Uganda (Acholi) regions, to manage the malaria resurgence that followed withdrawal of the intervention.” Please also note that several co-authors on this manuscript represent the Ugandan National Malaria Control Division and approve this terminology.

L198. Overall the evidence supporting the effectiveness of IRS in sub sahara Africa (either alone or in combination with LLIN) is rather limited... The authors forgot to talk about the Cochrane review of Pluess et al 2010 that showed limited evidence for supporting IRS for malaria control in stable malaria settings (Cochrane Review published in 2010). Please add a statement in the text.

Thank you for pointing us to this review. We have added a reference to it (see excerpted text below). However, we want to emphasize that this review focused on IRS in the absence of LLINs, while the present study evaluates the impact of IRS in the context of widely available LLINs.

“Thus, *while there is some evidence for the impact of IRS in the absence of LLINs*³⁰, it is surprising that the evidence base from contemporary controlled trials on the impact of adding IRS to LLINs for vector control is limited.”

L253. The authors should be more careful with the following statement “longer-acting formulations such as Actellic 300CS® administered every 2 years may be a cost-effective strategy for mitigating the risk of resurgence following sustained IRS and/or enabling countries to expand coverage when resources are limited).

First there’s no data on the residual activity of Actellic 300CS IRS in the paper so we don’t know for how long the treatment has remained effective in the study’s conditions (eg no residual bioassays on the walls were conducted by the team over the course of the study). Since the study was “observational (before-after and not a RCT) we cannot exclude the impact of other confounding factors on the outcomes (as highlighted by the authors in L261-263). Furthermore, they was no cost-assessment data provided in the paper so we can’t conclude on the cost effectiveness of IRS compared to other VC interventions (eg what is the cost for a single round of Actellic 300 per year in the study area?). Actually the higher cost of new formulated products compared to conventional ones can lead to a reduction of IRS coverage and then increase of disease incidence. The authors have to revise their statement considering the lack of data to support their claim.

Thank you for this valid point. We have changed the language in the discussion to imply that future research could formally assess this cost-effectiveness (with new text in Italics): “These data suggest that IRS with longer-acting formulations such as Actellic 300CS® administered every 2 years *could be considered as a strategy for mitigating the risk of*

resurgence following sustained IRS and/or enabling countries to expand coverage when resources are limited, *but formal assessment and a cost-effectiveness analyses are needed.*

Minor comments

Title: Can the authors replace “starting” by “re introducing” or “re starting” as IRS were previously implemented in all study sites.

Given the 5 sites included in Objective 3 did not receive IRS before it was initiated in 2014, we believe “starting” is appropriate.

Abstract. L46 Please precise the endpoints used to evaluate the impact of IRS.

We have added that the endpoints in this study are “changes in malaria incidence.”

Introduction;

L55. Update data from the WHO malaria report 2020

This has been updated throughout the text.

L62. New LLIN using synergist (Olyset Plus, Permanet 3.0) or non-pyrethroids (G2 combining alphacypermethrin and chlorfenapyr) have been developed and distributed in Africa to improve malaria control in pyrethroid resistance area and the authors should reflect that in the text.

We have now alluded to these nets in the introduction (new text in Italics): “Until recently, pyrethroids were the only class of insecticides approved for use in LLINs and, given the emergence of widespread pyrethroid resistance^{3,4}, there is concern that the effectiveness of LLINs may be diminishing, leading to the development of new LLIN formulations including pyrethroid synergists and non-pyrethroid nets.”

L76. Please indicate the insecticide and targeted dose used in IRS campaigns.

We provide a description of the insecticides used and timing of the IRS campaigns in the results section. More details of the specifics of these campaigns are available in the PMI Malaria Operational Plan for Uganda and this document has been added as a citation.

L228: resurgence “of malaria”

This has been edited.

REVIEWERS' COMMENTS

Reviewer #1 (Remarks to the Author):

I am happy with the responses and improvements to the manuscript.

Reviewer #2 (Remarks to the Author):

The authors addressed all my concern and I'm now satisfy with the revised version of the manuscript. The paper is suitable for publication in Nature Communications.